# JAK1 Signaling Is Involved in the Induction of Mechanical Alloknesis in Atopic Dermatitis

**DOI:** 10.3390/biomedicines13112744

**Published:** 2025-11-10

**Authors:** Ying Zuo, Sumika Toyama, Motoki Morita, Qiaofeng Zhao, Eriko Komiya, Soichiro Yoshikawa, Mitsutoshi Tominaga, Kenji Takamori

**Affiliations:** 1Juntendo Itch Research Center (JIRC), Institute for Environmental and Gender-Specific Medicine, Graduate School of Medicine, Juntendo University, 2-1-1 Tomioka, Urayasu-shi 279-0021, Japan; z.ying.cm@juntendo.ac.jp (Y.Z.); su-toyama@juntendo.ac.jp (S.T.); m.morita.wg@juntendo.ac.jp (M.M.); zhao@juntendo.ac.jp (Q.Z.); tominaga@juntendo.ac.jp (M.T.); 2Laboratory of Functional Morphology, Faculty of Pharmacy, Juntendo University, 6-8-1 Hinode, Urayasu-shi 279-0013, Japan; 3Department of Dermatology, Juntendo University Urayasu Hospital, 2-1-1 Tomioka, Urayasu-shi 279-0021, Japan

**Keywords:** alloknesis, atopic dermatitis, itch, JAK inhibitor

## Abstract

**Background/Objectives**: Mechanical alloknesis (m-alloknesis), the sensation of itch evoked by normally non-pruritic mechanical stimuli, is commonly observed in dry skin-associated conditions, such as xerosis, atopic dermatitis (AD), and psoriasis. Janus kinase (JAK) inhibitors are currently used to treat AD and suppress inflammation and itch. However, their specific roles in the modulation of m-alloknesis remain unclear. Therefore, in this study, we investigated the effects of various oral JAK inhibitors on m-alloknesis using a murine model of AD. **Methods**: An AD-like phenotype was induced in mice through the repeated topical application of an ointment containing *Dermatophagoides farinae* (house dust mite) extract. The mice were then orally treated with one of three JAK inhibitors: the JAK1/2 inhibitor baricitinib, the JAK1-selective inhibitor abrocitinib, or the JAK2-selective inhibitor AZ960. M-alloknesis was evaluated by quantifying scratching behavior in response to 30 controlled mechanical stimuli applied to lesional skin. **Results**: The JAK inhibitor treatments did not affect skin barrier integrity, dermatitis severity, or spontaneous scratching behavior. However, baricitinib and abrocitinib significantly reduced m-alloknesis scores, whereas AZ960 had no effect. **Conclusions**: These results suggest that JAK1 signaling plays a critical role in the induction of m-alloknesis in AD. Selective JAK1 inhibition is a promising therapeutic strategy for attenuating m-alloknesis and improving quality of life for patients with AD, independent of general skin inflammation and barrier function.

## 1. Introduction

Atopic dermatitis (AD) is a chronic inflammatory skin disorder characterized by recurrent episodes of exacerbation and remission [1,2]. It is clinically characterized by pruritus, erythema, xerosis, lichenification, and eczematous lesions that vary with age and disease severity. The condition typically follows a chronic and relapsing course and may be associated with other atopic disorders, such as asthma and allergic rhinitis [1,3]. Among the various symptoms of AD, pruritus is particularly challenging to manage. It is considered refractory in many cases because it is driven by a complex interplay of immune, neural, and barrier-related factors, making conventional antihistamines largely ineffective [4,5]. The pathogenesis of AD involves a complex interplay of genetic predisposition, epidermal barrier impairment, type 2 helper T cell (Th2)-dominant immune responses, and neuroimmune dysregulation. Cytokines, such as interleukin (IL)-4, IL-13, IL-31, and thymic stromal lymphopoietin (TSLP), are central to disease activity and act through the Janus kinase (JAK) signal transducer and activator of the transcription (STAT) pathway [6].

Although biologic therapies targeting IL-4/IL-13 (e.g., dupilumab) have significantly advanced the treatment of AD, many patients with moderate-to-severe disease have an incomplete response or contraindications to biologics [7]. As small-molecule agents, JAK inhibitors are characterized by a short half-life, rapid onset of action, and oral availability. Moreover, JAK inhibitors have exhibited high efficacy in providing more rapid relief than biologics, particularly with respect to reducing itch intensity [8,9,10]. Approved oral agents include baricitinib (JAK1/2), abrocitinib (JAK1-selective), and upadacitinib (JAK1-selective), while topical formulations, such as delgocitinib and ruxolitinib, provide alternatives for localized disease [6]. Furthermore, Eczema Area and Severity Index (EASI)-75 achievement rates were approximately 2.8-fold higher with oral JAK inhibitors than with a placebo. Although treatment-emergent adverse events slightly increased, the incidence of adverse events leading to treatment discontinuation was similar to that of the placebo group. Therefore, JAK inhibitors are considered an effective treatment option with a favorable safety profile [11,12,13]. Topical JAK inhibitors also achieved significant amelioration of skin inflammation and pruritus with a favorable safety profile, making them a promising non-systemic option for mild to moderate AD [14].

Although JAK inhibitors are very effective at alleviating skin symptoms and inflammation in AD patients, their efficacy against mechanical alloknesis (m-alloknesis) remains unclear. M-alloknesis refers to itch caused by harmless mechanical stimuli, which is transmitted via different neural pathways to itching caused by chemical stimuli (chemical allokinesis) [15]. Therefore, in this study, we investigated the effects of various oral JAK inhibitors on m-alloknesis in AD. We herein describe the JAKs involved in the induction of m-alloknesis and the efficacy of their inhibitors against m-alloknesis.

## 2. Materials and Methods

### 2.1. Animals

The NC/Nga strain shows innate atopic susceptibility and develops AD-like lesions, particularly under environmental stressors, and serves as a standard murine model for atopic dermatitis studies [16,17]. Male NC/Nga mice were purchased from Oriental Yeast (Tokyo, Japan) at 5–7 weeks of age. The mice were kept in house and used in experiments at 8–10 weeks of age. The mice were maintained in a controlled environment (22–24 °C) with a 12 h light/dark cycle. Standard food and tap water were available ad libitum. All animal experiments were conducted in accordance with institutional guidelines and received prior authorization from the Animal Ethics Committee at Juntendo University Graduate School of Medicine (Approval No. 2024104, 2025064, Approval Date: 5 March 2024, 28 February 2025, respectively).

### 2.2. Induction and Evaluation of AD-like Dermatitis

The protocol for AD induction was performed as previously described, with some modifications [18]. In brief, on day 0, hair was removed with an electric shaver under sevoflurane anesthesia, and residual hair was depilated using a hair removal cream. Two hours later, 100 μg of *Dermatophagoides farinae* body (Dfb) ointment (Biostir Inc. Kobe, Japan) was applied topically to the shaved area on each mouse. For the second and subsequent inductions, growing hair was removed with an electric shaver twice a week for 3 weeks, and the barrier was disrupted by applying 150 μL of 4% sodium dodecyl sulfate to shaved dorsal skin 2 h before the Dfb ointment was applied. To confirm the lesional skin condition, transepidermal water loss (TEWL) and hydration of the stratum corneum (SC) at the application sites were quantified using evaporimetry devices (a Tewameter^®^ TM210 and a Corneometer^®^ CM825 (Courage & Khazawa, Cologne, Germany), respectively). Dermatitis severity was scored based on four skin features—erythema/hemorrhage, dryness/scarring, edema, and excoriation/erosion—with each being graded according to their presence and severity from 0 to 3 (none, 0; mild, 1; moderate, 2; severe, 3). The clinical skin score was defined as the sum of individual scores and ranged from 0 to 12. Scratching behavior was measured 1, 2, and 3 weeks after the start of AD induction for 2 h using the SCLABA^®^-Next system (Noveltec, Kobe, Japan) (Figure 1a). 

### 2.3. Treatment of JAK Inhibitors

JAK inhibitors were administered in the same manner as previously reported [19]. In brief, we employed the following JAK inhibitors—baricitinib (MedChem Express, Monmouth Junction, NJ, USA) as a JAK1/2 inhibitor, abrocitinib (Selleckchem, Houston, TX, USA) as a JAK1-selective inhibitor, and AZ960 (Selleckchem) as a JAK2-selective inhibitor—at concentrations of 3, 15, and 1.58 mg/kg, respectively, based on the equivalence of effective concentrations selected according to IC50 value, as previously described [19]. Each JAK inhibitor was dissolved in dimethyl sulfoxide (DMSO, SIGMA, St. Louis, MO, USA) to a volume of 37.5 μL and adjusted to a volume of 1.5 mL with methyl cellulose (FUJIFILM Wako Pure Chemical Corporation, Osaka, Japan). The negative control was a 2.5% DMSO in a methyl cellulose solution (37.5 μL DMSO/1.4625 mL methyl cellulose). All JAK inhibitors and their negative controls were orally administered at 100 µL per 10 g of body weight per mouse on the day after generating AD mice (Figure 1a).

### 2.4. M-Alloknesis Assay

The m-alloknesis assay was carried out following established protocols [15]. At 1, 2, and 3 weeks after AD induction, the mice were transferred individually to clean cages and allowed to acclimate for ≥1 h. Non-noxious mechanical stimulation was applied to random points on the shaved upper back using von Frey filaments (0.07 or 0.16 g; Bioseb, Chaville, France). Each animal received three light touches at intervals exceeding 5 s (average 20 s), and this pattern was repeated 10 times over a 7.5 min session (total = 30 stimuli). The m-alloknesis score was defined as the cumulative number of scratching responses following stimulation. (Figure 1b). Partial blinding was maintained during the experiment and outcome assessment. While the treatment administrators remained unblinded by necessity, the behavioral experiments were conducted by labeling the groups with numerical codes. Additionally, the behavioral and dermatological assessors were kept unaware of each other’s findings to prevent mutual influence.

### 2.5. Statistical Analyses

Statistical analyses were conducted using GraphPad Prism 7 software (GraphPad Software, San Diego, CA, USA). The results are presented as mean ± SEM values. Group differences were assessed using one-way ANOVA with Sidak’s post hoc test and two-way ANOVA followed by Tukey’s multiple comparison test or the Kruskal–Wallis test with Dunn’s multiple comparison test, as appropriate.

## 3. Results

### 3.1. Short-Term Effects of JAK Inhibitors on Skin Inflammation in AD Model Mice

We focused on the short-term effects of JAK inhibitors on AD and administered several JAK inhibitors orally to a murine model after the induction of AD every week for three weeks. A single dose of JAK inhibitor during each week of AD induction did not significantly prevent the development of skin inflammation (Figure 2). Furthermore, we examined the effects of the JAK inhibitors on skin barrier function. Similarly, the single-dose oral administration of JAK inhibitors did not result in any detectable changes in skin barrier integrity (Figure 3).

### 3.2. Short-Term Effects of JAK Inhibitors on Itch in AD Model Mice

We next investigated the short-term effects of the JAK inhibitors on AD-induced pruritus. However, the single administration of the JAK inhibitors each week failed to suppress spontaneous scratching behavior in our AD model (Figure 4a). In contrast, although m-alloknesis associated with AD was not suppressed by the JAK2 inhibitor (AZ960), it was significantly suppressed by the JAK1 inhibitor (Abrocitinib) and JAK1/2 inhibitor (Baricitinib) (Figure 4b).

## 4. Discussion

JAK inhibitors are a class of therapeutic agents that suppress excessive immune responses and are widely used in the treatment of autoimmune diseases, such as rheumatoid arthritis, as well as inflammatory disorders, including psoriatic arthritis and ulcerative colitis [20,21,22]. Among their various effects, JAK inhibitors for AD have a rapid onset of action and high efficacy in attenuating itch and improving skin lesions. They are also effective in moderate to severe cases that are insufficiently controlled by conventional therapies. While attention to potential side effects is necessary, JAK inhibitors represent a highly promising treatment option [11,12,13,23,24]. However, the effects of JAK inhibitors on alloknesis in AD remain unknown; therefore, they were investigated in this study.

To investigate the potential mechanisms underlying their rapid antipruritic effects, we focused on the short-term effects of JAK inhibitors on itch behaviors in an AD model (Figure 1). The short-term administration of JAK inhibitors (a single dose per week) did not attenuate skin inflammation, barrier function, or spontaneous itch (Figure 2 and Figure 3). However, the short-term administration of JAK1-selective and JAK1/2 inhibitors, but not JAK2-selective inhibitors, suppressed m-alloknesis (Figure 4). These results suggest that m-alloknesis in AD is mediated through the JAK1 pathway. We previously reported that the short-term administration of JAK1-selective inhibitors suppressed m-alloknesis in a dry skin mouse model [19]. Furthermore, the administration of IL-4, IL-13, and TSLP induced m-alloknesis, which was suppressed by JAK1-selective inhibitors. Since we assume that TSLP expression is up-regulated in the lesional skin of AD, we suggest that m-alloknesis is triggered by TSLP released from keratinocytes upon scratching. Previous studies also showed that TSLP activated type 2 innate lymphoid cells and Th2, thereby promoting the secretion of IL-4 and IL-13. These type 2 cytokines, in turn, bind to their receptors on peripheral nerves, activate the JAK signaling pathway, and contribute to itch hypersensitivity [19,25,26]. Dry skin is a precursor symptom of AD, and AD exhibits a similar or more robust type 2 inflammatory profile than dry skin. Therefore, these findings indicate that m-alloknesis in AD is mediated by JAK1 signaling, potentially triggered by the binding of type 2 cytokines, such as TSLP, IL-4, and IL-13, which activate corresponding receptors on immune cells and sensory neurons, leading to itch sensitization [27,28]. 

To date, histamine, bovine adrenal medulla (BAM) 8–22, 5-hydroxytryptamine (serotonin), and protease-activated receptor (PAR) 4 agonists have been shown to induce m-alloknesis, while the μ-opioid receptor antagonist naltrexone and the κ-opioid receptor agonist nalfurafine suppressed histamine-induced m-allokinesis [29]. The aryl hydrocarbon receptor, which is a nuclear receptor that modulates responses to environmental stimuli, regulates the expression of artemin and induces thermal alloknesis. Moreover, the enzyme 11b-hydroxysteroid dehydrogenase-1, which is expressed in keratinocytes, suppressed alloknesis through artemin [30,31]. Collectively, these findings suggest the involvement of both peripheral and central mechanisms in the modulation of alloknesis. In the peripheral nervous system, mediators, such as type 2 cytokines, histamine, BAM8-22, serotonin, and PAR4, are considered to sensitize sensory neurons to mechanical stimuli. In parallel, aryl hydrocarbon receptors and type 2 cytokines within the central nervous system may also contribute to alloknesis. In this study, the incomplete suppression of alloknesis by JAK1 inhibitors implies the involvement of additional type 2 cytokine-independent pathways.

In clinical settings, JAK inhibitors are generally administered on a daily basis, and repeated daily dosing may lead to cumulative effects that significantly ameliorate skin inflammation, enhance barrier function, and reduce itch [32,33]. Notably, the results of this study show that even a single dose of the JAK1 inhibitor resulted in a rapid and marked reduction in m-alloknesis in the AD model but did not markedly attenuate skin inflammation or improve barrier function. This implies that the fast suppressive effect of the JAK1 inhibitor on mechanical itch hypersensitivity in AD may be independent of its classical anti-inflammatory or barrier-restoring properties. This finding is consistent with recent clinical trials and meta-analyses comparing the efficacy of various targeted systemic therapies in patients with atopic dermatitis, which identified JAK1-selective inhibitors as the most effective agents for alleviating pruritus compared with biologics and other types of JAK inhibitors. Among the JAK1-selective inhibitors, upadacitinib demonstrated superior efficacy over abrocitinib in itch reduction. Therefore, future studies are needed to explore the potentially distinct effects of JAK1 inhibition on mechanical alloknesis [34,35]. However, in this present study, the recording of spontaneous scratching was conducted after the alloknesis assay, which may have delayed the assessment beyond the optimal time for the efficacy of JAK inhibitors on spontaneous itch behavior. A slight reduction was noted in spontaneous scratching following JAK inhibitor administration in the second week after the induction of AD. These results indicate the potential antipruritic effects of these JAK inhibitors.

## 5. Conclusions

The results of this study reveal the fast and significant suppressive effects of JAK inhibitors on m-alloknesis, suggesting the involvement of JAK1 signaling in the regulation of itch sensitization in AD.

## Figures and Tables

**Figure 1 biomedicines-13-02744-f001:**
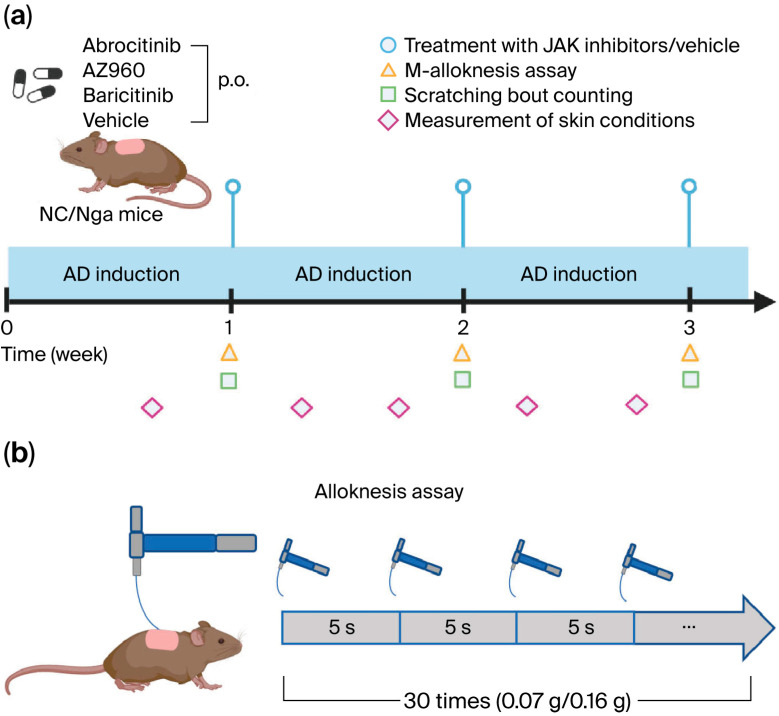
Experimental scheme. (**a**) Schema of the procedure to establish the AD mouse model and treatments with various JAK inhibitors or vehicles. (**b**) Schema of the procedure to assess m-alloknesis. p.o.: per os. Created in BioRender. ZUO, Y. (2025) https://BioRender.com/mm7pvv3 (accessed on 9 November 2025).

**Figure 2 biomedicines-13-02744-f002:**
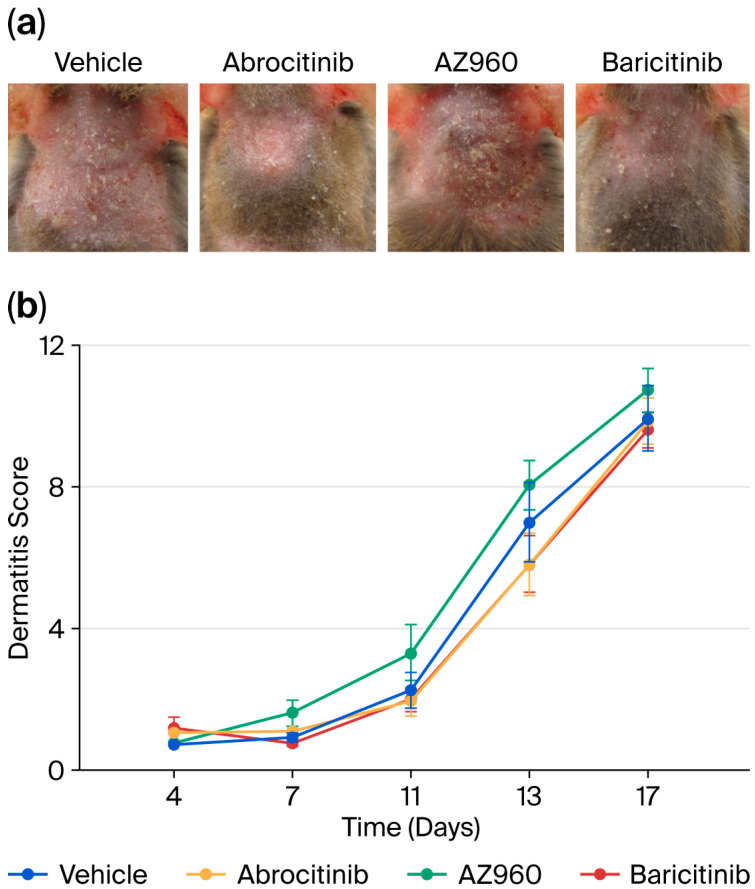
Short-term effects of various JAK inhibitors on skin inflammation in AD mice. (**a**) Representative images of AD-like lesions in the dorsal skin of NC/Nga mice treated with various JAK inhibitors (Day 17). (**b**) Dermatitis scores used to evaluate the severity of dorsal skin inflammation in our AD mouse model. *n* = 8 for each group. Mean ± SEM for each group.

**Figure 3 biomedicines-13-02744-f003:**
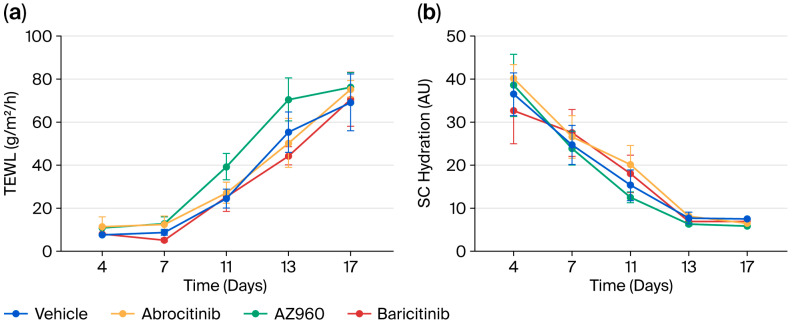
Short-term effects of various JAK inhibitors on the skin barrier function of AD mice. (**a**) Changes in TEWL during the experiment period. (**b**) Changes in SC hydration during the experiment period. *n* = 8 for each group. Mean ± SEM for each group.

**Figure 4 biomedicines-13-02744-f004:**
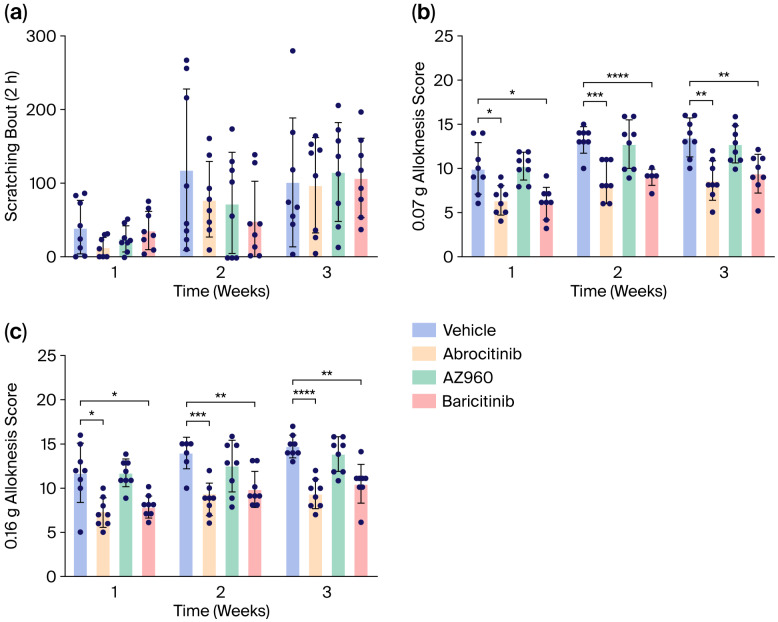
Short-term effects of various JAK inhibitors on scratching behaviors and m-alloknesis in AD mice. (**a**) Number of spontaneous scratching bouts in AD mice treated with various JAK inhibitors (scratching behaviors were recorded for 2 h). (**b**,**c**) Short-term effects of various JAK inhibitors on m-alloknesis by von Frey filaments of 0.07 g (**b**) and 0.16 g (**c**). *n* = 8 for each group. Mean ± SEM for each group. * *p* < 0.05; ** *p* < 0.01; *** *p* < 0.001; **** *p* < 0.0001.

## Data Availability

All data are contained within the manuscript.

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
