# Peer review of "JAK1 Signaling Is Involved in the Induction of Mechanical Alloknesis in Atopic Dermatitis"

_biomedicines, 2025, doi:10.3390/biomedicines13112744_

Round 1

Reviewer 1 Report

Comments and Suggestions for Authors

As a distressing yet poorly understood symptom, alloknesis is likely present in many other conditions and may impose a significant burden. This study aimed to investigate the therapeutic effects of oral JAK inhibitors on mechanical alloknesis in atopic dermatitis (AD). The findings demonstrate that JAK1 signaling plays a critical role in the activation of mechanical alloknesis, suggesting the potential efficacy of JAK1 inhibitors in treating alloknesis-related disorders.

However, several issues need to be addressed:

  1. The scoring criteria for dermatitis in Figure 2b should be clearly defined.
  2. Please explain why the JAK2 inhibitor AZ960 appears to exacerbate inflammation and mechanical alloknesis compared to the vehicle group.
  1. Given the hypothesis that TSLP induces mechanical alloknesis in the AD model, the expression of TSLP in lesional skin following different treatments should be evaluated.

Author Response

Reviewer 1

As a distressing yet poorly understood symptom, alloknesis is likely present in many other conditions and may impose a significant burden. This study aimed to investigate the therapeutic effects of oral JAK inhibitors on mechanical alloknesis in atopic dermatitis (AD). The findings demonstrate that JAK1 signaling plays a critical role in the activation of mechanical alloknesis, suggesting the potential efficacy of JAK1 inhibitors in treating alloknesis-related disorders. However, several issues need to be addressed:

Comment 1: The scoring criteria for dermatitis in Figure 2b should be clearly defined.

Response 1:

Thank you for your comment. This point has been clarified in Section 2.2. “Induction and evaluation of AD-like dermatitis,” in the Materials and Methods section:

Page 3 line 9599:

“The severity of dermatitis was assessed based on four symptoms: erythema/hemorrhage, scarring/dryness, edema, and excoriation/erosion. Each symptom was graded according to its presence and severity on a scale from 0 to 3 (none, 0; mild, 1; moderate, 2; severe, 3). The clinical skin score was defined as the sum of the individual scores, yielding a total score ranging from 0 to 12.”

Comment 2: Please explain why the JAK2 inhibitor AZ960 appears to exacerbate inflammation and mechanical alloknesis compared to the vehicle group.

Response 2:

Thank you for your comment. The JAK2 inhibitor group showed a slightly higher dermatitis score, which may be attributed to individual variability in inflammatory responses among mice, even under identical AD induction conditions. However, there were no statistically significant differences in dermatitis scores or skin barrier parameters between the groups. In addition, as shown in Figure 4b and 4c, the alloknesis scores were comparable between the AZ960 and vehicle groups. Therefore, we consider the minor increase in inflammation to result from individual differences in inflammatory response rather than any specific effect of AZ960.

Comment 3: Given the hypothesis that TSLP induces mechanical alloknesis in the AD model, the expression of TSLP in lesional skin following different treatments should be evaluated.

Response 3:

Thank you for your valuable comment. We agree that examining the expression of TSLP and related cytokines would provide important insights into the mechanisms driving mechanical alloknesis in AD. Type 2 cytokines such as IL-4, IL-13, IL-31, and TSLP are known to activate JAK kinases, initiating inflammatory and pruritogenic processes that contribute to alloknesis. JAK inhibitors can suppress this cascade by blocking downstream signaling. In our discussion, we have addressed these potential mechanisms.

Moreover, in our previous study (Toyosawa et al., 2025; DOI: 10.1016/j.jdermsci.2024.10.002), we demonstrated that not only TSLP but also IL-4 and IL-13 can induce mechanical alloknesis in mice. These findings support our hypothesis that JAK inhibitors alleviate alloknesis by inhibiting JAK-mediated signaling of these itch-related cytokines.

At present, we have not directly measured changes in TSLP expression following JAK inhibitor treatment. However, since JAK functions downstream of these cytokines, we consider it unlikely that JAK inhibition would substantially alter cytokine expression levels themselves. We appreciate your thoughtful suggestion and intend to investigate this aspect in future studies.

Reviewer 2 Report

Comments and Suggestions for Authors

The article presents intriguing and innovative research results dealing with the role of JAK pathways in alloknesis and possible therapeutic properties of JAK inhibitors in the modulation of that kind of itch.

The introduction provides a comprehensive overview of the topic of the research and indicates in an understandable way the justification of tackling on the subject. The authors provide a detailed description of methods allowing for the experiments to be repeated, if desired. The results indicate convincingly the role of baricitinib and abrocitinib upon the reduction of the alloknesis scores in the employed mouse model.

I would encourage the Authors to extend their discussion and provide justification for the choice of BARI, ABRO and AZ960 JAK-inhibitors while not attempting to assess upadacitinib, which is known for its potent anti-pruritic (among others) activities in AD treatment. This property of upadicitinib has been proven not only in clinical trials but also in meta-analysis of studies showing comparisons with other available treatment modalities, including biologics, e.g., Dermatol Ther (Heidelb) (2022) 12:1181–1196, https://doi.org/10.1007/s13555-022-00721-1. This may be an interesting addendum to the discussion of these interesting results. Moreover, these meta-analyses have been widely present in current scientific discussions and debates about AD management. Although the study touches upon mouse model, the attempt to extrapolate the results on the clinical real-world practice might be envisaged and touched upon during interpretation fo the results.

Otherwise, I have no issues to be raised with regard to this manuscript. This is an interesting results with possible implications for the allergy/dermatology practice.

Author Response

The article presents intriguing and innovative research results dealing with the role of JAK pathways in alloknesis and possible therapeutic properties of JAK inhibitors in the modulation of that kind of itch.

The introduction provides a comprehensive overview of the topic of the research and indicates in an understandable way the justification of tackling on the subject. The authors provide a detailed description of methods allowing for the experiments to be repeated, if desired. The results indicate convincingly the role of baricitinib and abrocitinib upon the reduction of the alloknesis scores in the employed mouse model.

I would encourage the Authors to extend their discussion and provide justification for the choice of BARI, ABRO and AZ960 JAK-inhibitors while not attempting to assess upadacitinib, which is known for its potent anti-pruritic (among others) activities in AD treatment. This property of upadacitinib has been proven not only in clinical trials but also in meta-analysis of studies showing comparisons with other available treatment modalities, including biologics, e.g., Dermatol Ther (Heidelb) (2022) 12:1181–1196, https://doi.org/10.1007/s13555-022-00721-1. This may be an interesting addendum to the discussion of these interesting results. Moreover, these meta-analyses have been widely present in current scientific discussions and debates about AD management. Although the study touches upon mouse model, the attempt to extrapolate the results on the clinical real-world practice might be envisaged and touched upon during interpretation of the results.

Otherwise, I have no issues to be raised with regard to this manuscript. This is an interesting results with possible implications for the allergy/dermatology practice.

Response:

Thank you very much for your thoughtful and encouraging comments. We are pleased to note that the clinical evidence aligns well with our findings. As reported in the meta-analysis by Heidelb et al. (Dermatol Ther (Heidelb), 2022), in terms of itch relief (ΔNRS ≥ 4), the three most effective treatments were all JAK1-selective inhibitors: high-dose upadacitinib, high-dose abrocitinib, and low-dose upadacitinib. Their antipruritic efficacy surpassed that of biologics and non–JAK1-selective inhibitors, with upadacitinib appearing to be the most effective in relieving itch in AD patients.

In our study, however, we selected one representative inhibitor from each JAK subtype for comparison and did not specifically examine differences among JAK1-selective inhibitors. Moreover, our research focused on the short-term effects of JAK inhibition on mechanical alloknesis in a mouse model, which may not directly reflect the broader or longer-term antipruritic mechanisms observed in clinical settings. Therefore, direct comparison with clinical trial data is limited. However, we agree that it would be highly valuable to evaluate the effects of upadacitinib on mechanical alloknesis in our experimental model in future studies. We appreciate your suggestion to highlight this clinical perspective and have revised the Discussion accordingly, as follows:

Page 7 line 226232:

“This finding is consistent with recent clinical trials and meta-analyses comparing the efficacy of various targeted systemic therapies in patients with atopic dermatitis, which identified JAK1-selective inhibitors as the most effective agents for alleviating pruritus compared with biologics and other JAK inhibitors. Among the JAK1-selective inhibitors, upadacitinib demonstrated superior efficacy over abrocitinib in itch reduction. Therefore, future studies are warranted to explore the potentially distinct effects of JAK1 inhibition on mechanical alloknesis.”

Round 2

Reviewer 1 Report

Comments and Suggestions for Authors

Accept